# The effect of sodium carboxymethyl starch with high degree of substitution on defecation

**Wu-dang Lu**[☯]**, Man-li Wu**[☯]**, Jun-xia Zhang, Ting-ting Huang, Shuai-shuai Du, Yong-xiao Cao***

School of Basic Medical Sciences, Xi'an Jiaotong University Health Science Center, Xi'an, Shaanxi, China

☯ These authors contributed equally to this work.
* yxy@xjtu.edu.cn

**Data Availability Statement:** All relevant data are within the manuscript.

## Abstract

Sodium carboxymethyl starch (CMS-Na), a kind of food additive with high degree of substitution, is also known as a prebiotic. The aim of this study was to determine the effect of CMS-Na on defecation. Constipated mouse model was prepared by loperamide. Normal rats were also used in the study. Short-chain fatty acids in rat feces were detected by gas chromatography. The bacterial communities in rat feces were identified by 16S rDNA gene sequencing. 5-hydroxytryptamine (5-HT) and tryptophan hydroxylase 1 (Tph1) were measured by ELISA. The results showed that CMS-Na increased the fecal granule counts and intestinal propulsion rate in constipated mice. The contents of water, acetic acid, propionic acid and n-butyrate in feces, Tph1 in colon and 5-HT in serum of rats were increased. In addition, CMS-Na shortened the colonic transport time in rats. The 16S rDNA gene sequencing results indicated that CMS-Na increased the relative abundance of *Alloprevotella* and decreased the proportion of *Lactobacillus*. However, the biodiversity of the normal intestinal flora was not altered. In conclusion, CMS-Na can promote defecation in constipated mice. The mechanism may be related to the regulation of *Alloprevotella* and *Lactobacillus* in colon, the increase of short-chain fatty acids, and the promotion of the synthesis of Tph1 and 5-HT.

## Introduction

Functional constipation is a common gastrointestinal disorder in elderly persons and children; and affects 12% -17% of the general population worldwide. Constipation for a long time can cause hemorrhoids, endocrine disorders, anal fissure and even bowel cancer, which has seriously affected human health and life quality [1, 2]. Over the past few decades, osmotic and stimulant laxatives have been used to treat constipation. But some unwanted effects, such as artery contraction, coronary spasms and myocardial infarction, may occur. Therefore, it is necessary to study drugs to relieve constipation [3–5].

Dietary changes are one of the main causes of functional constipation, prebiotics are increasingly seen as functional healthy foods to alleviate constipation, which cannot be digested in the body, and play an important roles in balancing many key functions of the

**Funding:** This research was financially supported by Lipont Pharmaceutical Co. (China). The funders had no role in study design, data collection and analysis, decision to publish, or preparation of the manuscript.

**Competing interests:** The research was financially supported by Lipont Pharmaceutical Co. The study is not related to employment, consultancy, patents, products in development, marketed products of supporter. The funders had no role in study design, data collection and analysis, decision to publish, or preparation of the manuscript. we declare that there is no competing conflict of interest. This does not alter our adherence to PLOS ONE policies on sharing data and materials.

intestinal flora [6–8]. Prebiotics include fructose oligosaccharides, cellulose and resistant starch, etc. Sodium carboxymethyl starch (CMS-Na) is an important resistant starch which resists amylase hydrolysis *in vitro*. It is not hydrolyzed to D-glucose in the small intestine, but can be fermented in the colon by anaerobic bacteria [9, 10]. CMS-Na is a plant polysaccharide (plant starch) extracted from maize that is modified by carboxymethyl to become etherified starch sodium salt (modified starch sodium salt), as shown in Fig 1. When the substitution degree is greater than 0.2, CMS-Na can be prepared into oral solution to enhance immunity [11, 12]. Here, we investigated the effect and mechanism of CMS-Na in constipation, we speculated that CMS-Na could increase the frequency of defecation in clinical application.

## Materials and methods

### Animals

Kunming mice weighing 18-22g and Sprague-Dawley rats weighing 200-220g (quality certificate No.SCXK 2012–003) were used in this study. The temperature of animal house was $18\sim26°C$, and the humidity was $40\%\sim70\%$.

All animals were purchased from the Animal Center of Xi'an Jiaotong University (Shaanxi, China) and were conducted in compliance with the guidelines of ethical animal research. The study protocol was approved by the Ethics Committee of Xi'an Jiaotong University Health Science Center. The animals were sacrificed by sodium pentobarbital after experiment.

### Drugs

CMS-Na with the substitution degree of 0.5, mosapride, metronidazole, and propofol were provided by Lipont Pharmaceutical Co., Ltd., Xi'an, Shaanxi, China. Loperamide was manufactured by Janssen Pharmaceutical Co., Ltd. Xi'an, Shaanxi, China.

### Constipation mouse model and defecation measurement

After fasting for 12 h, the mice were given loperamide (ig, 10 mg $kg^{-1}$) for 3 days to establish constipation model. Constipated mice were randomly divided into five groups: ① the model group, ②~④ the CMS-Na (0.45, 0.9, 1.8 g $kg^{-1}$) groups, and ⑤ the mosapride group (0.2 mg $kg^{-1}$). Normal mice were used as normal control. Each group included 12 mice. CMS-Na or

**Fig 1. Sodium carboxymethyl starch (CMS-Na) is prepared by reaction from plant starch.**

mosapride were given once a day for 3 days, respectively, and then the fecal granules of mice were recorded for 3 h. Afterwards, the mice were fasted for 12 h. After treatment with CMS-Na or mosapride for 1.5 h, mice were gavaged with red pitaya juice (20 mL kg$^{-1}$). Thirty minutes later, the small intestine of mice was separated. The total length of the intestine and the intestinal propulsion length from the pylorus to the front of the red intestinal content were measured. The intestinal propulsion rate was calculated according to the following formula:

Intestinal propulsive rate (%) = intestinal propulsive length/total intestinal length × 100%.

## Normal rats and defecation measurement

Rats were randomly divided into 5 groups with 10 rats in each group, including ①the normal group, ②~④ the CMS-Na groups (0.3, 0.6 and 1.2 g kg$^{-1}$), and ⑤ the mosapride group (1.2 mg kg$^{-1}$). CMS-Na or mosapride were given once a day for 7 d. At first, the feces were collected and weighed. The water content of feces was calculated according to the following formula:

$$\text{Fecal moisture (\%)} = (\text{wet weight} - \text{dry weight})/\text{wet weight} \times 100\%$$

Then, the rats were anesthetized with propofol (10 mg kg$^{-1}$) intraperitoneally, push a glass ball with 3.5 mm in diameter into the anus for about 6 cm at the end of the colon. Rats were separated in single cages, and the time to discharge the glass pellet was recorded. About 100 mg of feces was collected from each rat and thoroughly mixed with 10 mL of 0.9% NaCl. The pH of all samples was measured.

## Measurement of short-chain fatty acids and 16S rDNA sequence in feces

To investigate the impact of CMS-Na on short-chain fatty acids (SCFAs), rats were randomly divided into normal group and CMS-Na 1.2 g kg$^{-1}$ group with 10 rats in each group. Rats were treated with CMS-Na once a day for 5 d. Feces (200 mg) was collected and thoroughly mixed with 2 mL of water. The samples were centrifuged at 4000 rpm for 10 min. The supernatant (0.8 mL) was added to 0.2 mL of 4 M HCl and 1mL of an ether solution. The mixture was centrifuged for 10 min at 4000 r min$^{-1}$, and the upper ether phase was used to determine the molar profile of SCFAs by gas chromatography (Trace GC UItra, Thermo, USA) equipped with a hydrogen flame ionization detector (FID) and a polar capillary column of WM−FFAP (30 m× 0.25 mm × 0.25 μm). The GC was set as follows: carrier gas, nitrogen; column flow rate, 2.0 mL min$^{-1}$; the shunt ratio, 5:1; air flow, 300 mL·min$^{-1}$; hydrogen flow, 40 mL min$^{-1}$; and tail blowing (nitrogen) flow, 30 mL min$^{-1}$. In brief, the supernatant (1 μL) was injected at 220˚C. The column temperature was 35˚C at the time of injection and was increased by 5˚C every minute until reaching 230˚C and then held for 5 min. The detector temperature was 230˚C. Taking acetic acid, propionic acid and butyric acid as standard solutions and using ether as the solvent, a series of mixed standard concentrations of 100, 50, 25, 20, 10 and 5 μg mL$^{-1}$ were prepared and analyzed. The standard curve was drawn with the concentration of the standard substance and the corresponding peak area ratio as ordinates.

Stool samples from 10 rats per group were collected in a sterile microtube, frozen in liquid nitrogen, and stored in a −80˚C freezer until analysis. Total DNA was isolated using a Wizard Genomic DNA purification kit following the manufacturer's instructions (Promega, Madison, USA). The relative abundance of bacteria was measured using 16S rDNA lysis. The V4 regions of the bacterial 16S rDNA gene were amplified by polymerase chain reaction. The raw data from the 16S rDNA gene sequencing were organized into operational taxonomic units at 97% identity using UPARSE [13]. The taxonomy was analyzed by the Ribosomal Database Project and used as a reference database. The α-diversity represented the analysis of diversity data, including Chao 1, the Simpson index, and the Shannon index [14].

### Analysis of colonic Tph1 and serum 5-HT

Rats were randomly divided into 4 groups with 10 rats in each group: ① the normal group, ② the CMS-Na group (1.2 g kg⁻¹), ③ the CMS-Na + metronidazole group (1.2 g kg⁻¹ + 0.3 g kg⁻¹), and ④ the CMS-Na + 4-chloro-dl-phenylalanine group (1.2 g kg⁻¹ + 10 mg kg⁻¹. Rats were orally treated with drugs once a day for 5 d. On the last day, rats were fasted for 24 h. Rats were anesthetized, and blood was taken. ELISA was used to assess 5-hydroxytryptamine (5-HT) in serum according to the manufacturer's instructions. The proximal colon (about 400 mg) was removed and cut into small pieces. The pieces were added to 4 mL of 1 M PBS and put in a high flux refrigeration centrifuge for 5 minutes at 40 Hz, 4°C. The homogenates were stored overnight at -20°C. After two freeze-thaw cycles to break the cell membranes, the homogenates were centrifuged for 10 min at 4000 r min⁻¹; the supernatant was analyzed by ELISA according to the manufacturer's instructions. The contents of tryptophan hydroxylase 1(Tph1) and 5-HT were investigated.

### Statistical analysis

Statistical analysis was performed using *SPSS* version 18.0. All data are presented as the mean ±standard deviation (SD), and a *P*<0.05 was considered statistically significant. Data normality was assessed using the Shapiro-Wilk test. ANOVA with least significant difference was used to analyze quantitative data.

## Results

### Effect of CMS-Na on defecation in constipated mice

As shown in Fig 2, the total fecal grains and intestinal propulsion rate in constipation model mice decreased obviously compared with the normal mice, which suggesting the constipation model was successful. The number of fecal grains of the CMS-Na groups (0.45, 0.9 and 1.8 g kg⁻¹) significantly increased (*P*<0.01) compared with the model group. The number of fecal grains in the 1.8 g kg⁻¹ group (22.9±8.4) were higher than those in the mosapride group (14.3 ±8.4) (*P*<0.05). The intestinal propulsive rate of the CMS-Na groups (75.3%±9.2%, 78.7% ±9.6%, 79.4%±8.0%) significantly increased compared with the model group (63.0%±13.6%) (*P*<0.05 or *P* <0.01). The intestinal propulsive rate of the 1.8 g kg⁻¹ group was higher than that of the mosapride group (70.9%±8.5%) (*P*<0.05). Therefore, CMS-Na can relieve constipation in constipated mice.

### Effect on colonic transit time on normal rats

The colonic transit time in normal rats was 148 ± 47 min. After 7 d of CMS-Na (0.6 g·kg⁻¹ and 1.2 g kg⁻¹) treatments, the rat colonic transit time decreased to 93 ± 41 min, and 98 ± 50 min respectively (*P*<0.01, *P*<0.05). The colonic transit time of the mosapride group was decreased (109±35 min) (Fig 3). Although there was a decreasing trend for the colonic transit time in the CMS-Na 0.3 g kg⁻¹ group, there was no significant difference in the colonic transit time between the CMS-Na 0.3 g kg⁻¹ group and the normal group.

### Effect on Fecal water content and pH of feces of normal rats

The results in Fig 4 showed that the fecal water content (56.0%±4.1%, 57.7%±10.4%, 60.2% ±6.8%) of the CMS-Na group (0.3 g kg⁻¹, 0.6 g kg⁻¹ and 1.2 g·kg⁻¹) significantly increased (*P*<0.05 or *P*<0.01) compared with the normal group (49.0%±6.8%), and had a positive correlation to dosage. It is suggested that CMS-Na can improve the fecal moisture content of rats. The fecal pH of the CMS-Na groups (0.3 g·kg⁻¹, 0.6 g·kg⁻¹ and 1.2 g·kg⁻¹) was significantly

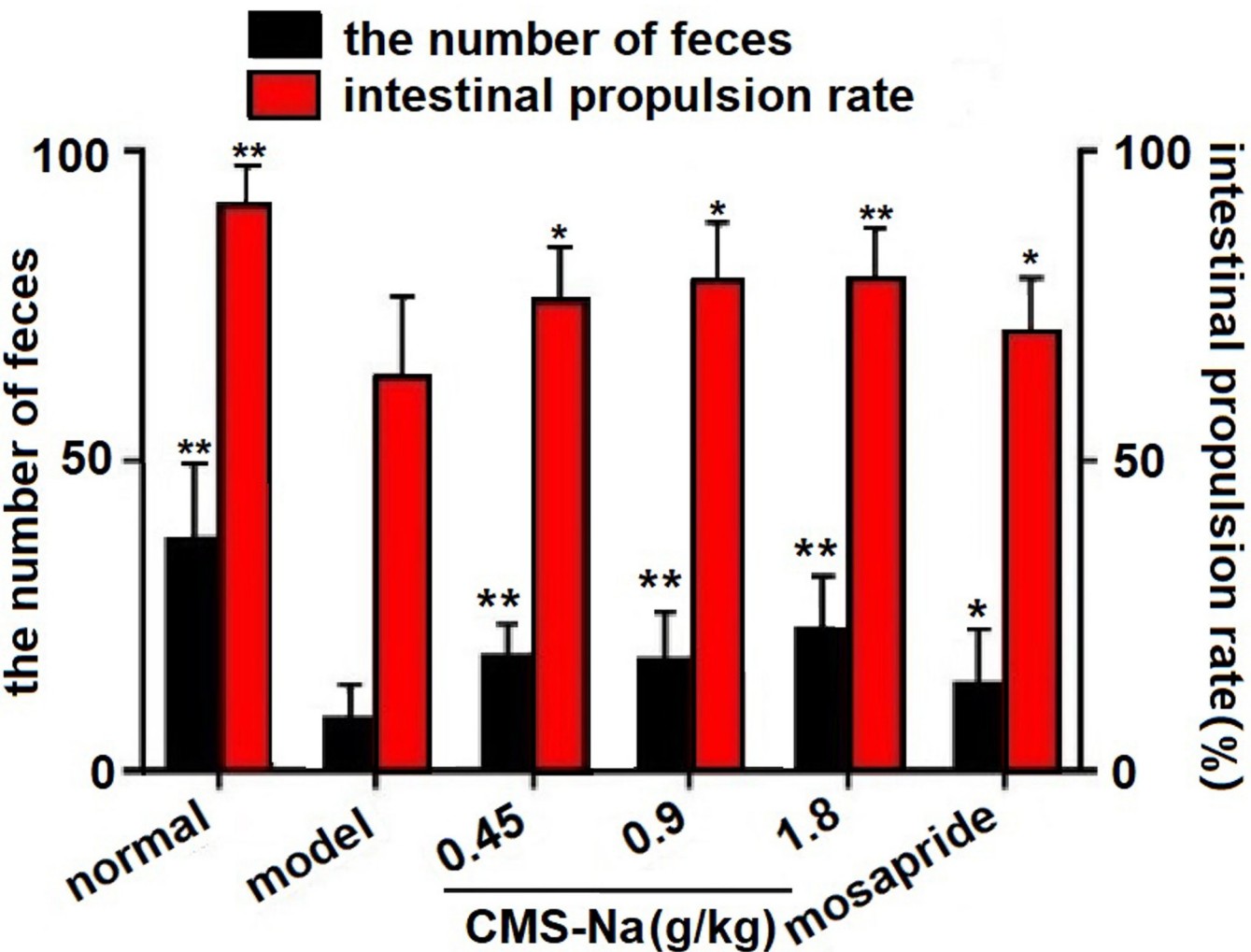

**Fig 2. Effect of CMS-Na on defecation and intestinal propulsion in loperamide-induced constipated mice.** Mice were given (ig) CMS-Na or mosapride after 3 h of loperamide once a day for 3 d. The fecal grains of mice were recorded for 3 h. After being fasting for 12 h, mice were gavaged with CMS-Na or mosapride. One and half hours later, mice were gavaged with red pitaya juice. Half an hour later, the small intestine was separated. The total length of the intestine and the intestinal propulsion length from the pylorus to the front of the red intestinal content were measured. Mean±SD, $n$ = 12, $^*P<0.05$, $^{**}P<0.01$ vs. model; $^\#P<0.05$ vs. mosapride.

lower than that of the normal group (6.67±0.14, 6.51±0.08, 6.41±0.08 vs. 6.84±0.14; $P<0.05$ or $P<0.01$), suggesting that CMS-NA can reduce the intestinal pH of normal rats.

## Content of short-chain fatty acids in feces of normal rats

It is presumed that intestinal microorganisms fermenting CMS-Na produces short-chain fatty acids (SCFAs) and lows intestinal pH. The contents of acetic acid, propionic acid and n-butyric acid of the feces in CMS-Na 1.2 g·kg$^{-1}$ group significantly increased compared with the normal group (13.71±2.4 μmol/g vs.1.58±0.24 μmol/g, 7.38±1.28 μmol/g vs.0.21±0.21 μmol/g, 11.37±2.09 μmol/g vs.0.15±0.15 μmol/g) ($P <0.01$, Fig 5).

## Microbial diversity and composition in feces of normal rats

Ten sample sequences were classified by operational taxonomic units. There were no significant differences in the diversity and richness of the gut microbiota between the CMS-Na and

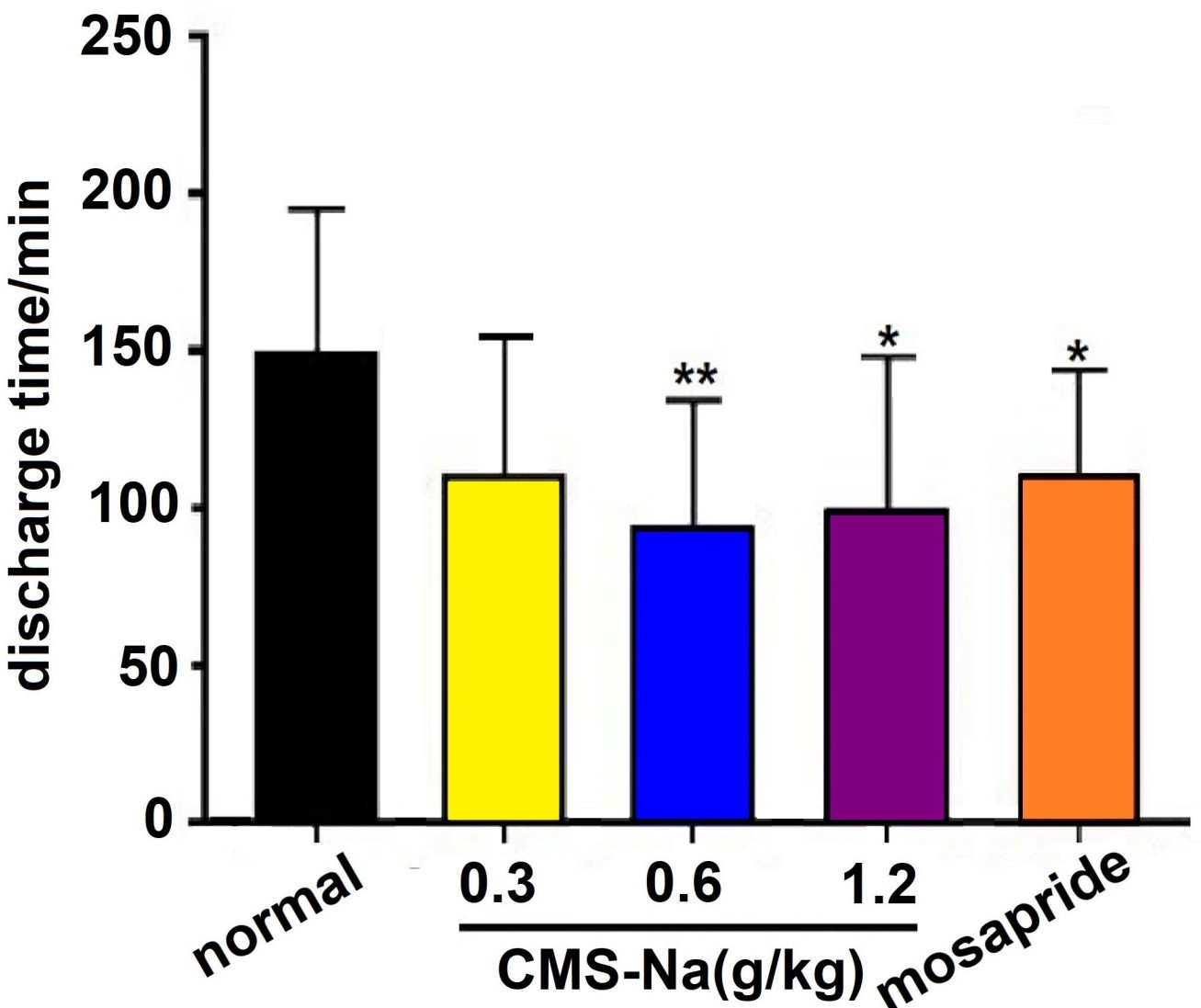

**Fig 3. Effect of CMS-Na on colonic transit in normal rats.** Rats were orally treated with 0.3, 0.6, or 1.2 g·kg$^{-1}$ CMS-Na or 1.2 mg·kg$^{-1}$ mosapride once a day for 7 d. Then, 3.5 mm glass beads were inserted into the junction of the rectum and the colon in rats and the discharge time was recorded. Mean±SD, $n$ = 10, $^*P$<0.05,$^{**}P$<0.01, vs. normal.

normal groups ($n$ = 10, $P$>0.05). The diversity of the microbial composition was analyzed at the *phylum* and *genus* levels. At the *phylum* level, the abundance of *Bacteroidetes* and *Proteobacteria* was not significantly altered in both CMS-Na 1.2 g·kg$^{-1}$ groups, nor were *Firmicutes* or *Actinobacteria* between the CMS-Na 1.2 g·kg$^{-1}$ group and normal group (Fig 6). Fig 7 shows that at the *genus* level, the abundance of beneficial bacteria changed. *Lactobacillus* significantly decreased (CMS-Na, 2.8%±.6% vs. normal, 8.1%±4.5%, $P$<0.01), but significantly *Alloprevotella* increased (CMS-Na, 5.5%±5.3% vs. normal, 1.2%±1.4%, $P$<0.05).

## Colonic Tph1 and 5-HT in serum of normal rats

To further determine the effect of CMS-Na on the expression of SCFAs, the Tph1 levels in the proximal colon and 5-HT in serum were measured (Fig 8). Compared with the normal group,

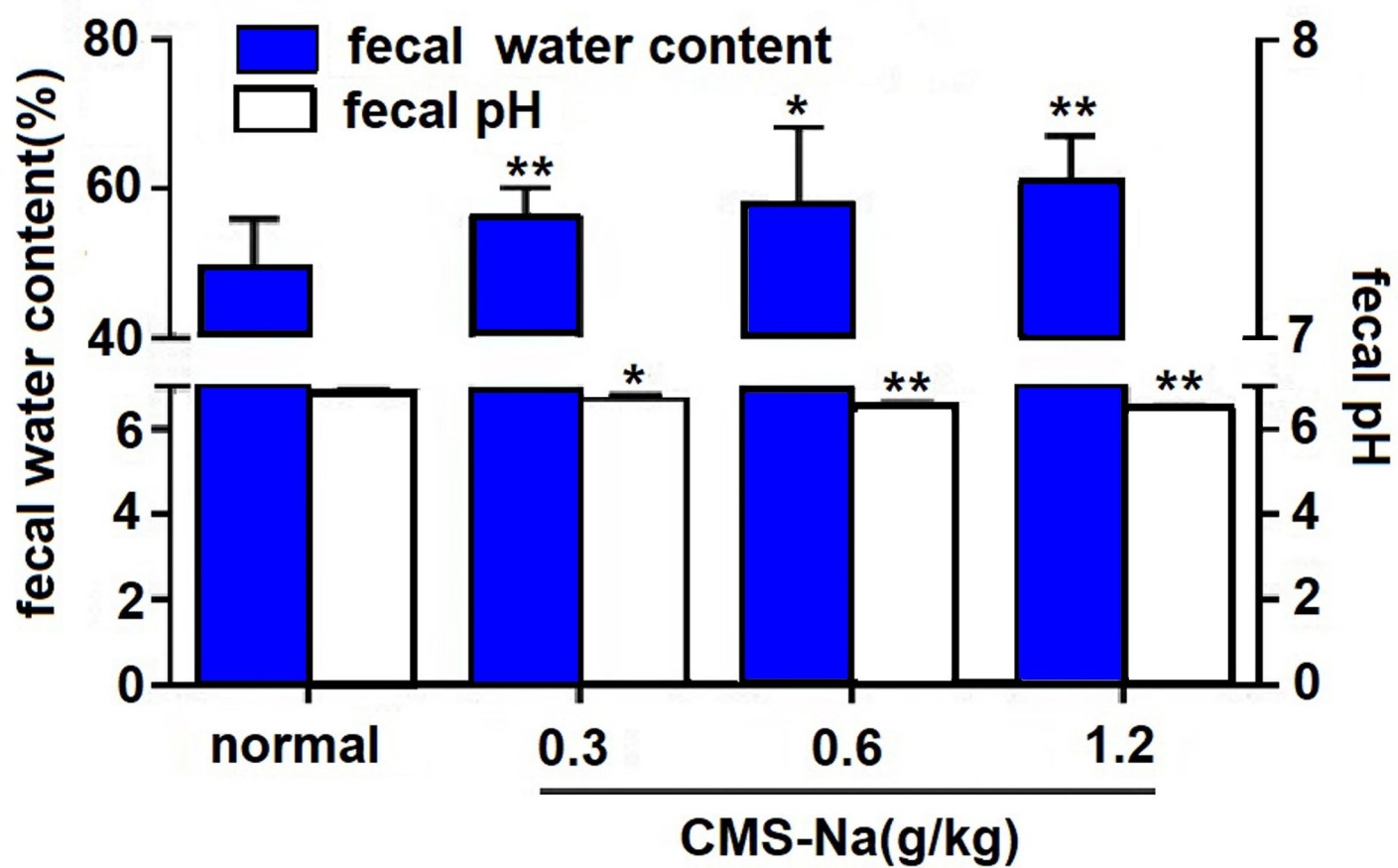

**Fig 4. Effect of CMS-Na on the fecal water content and fecal pH of rats.** Rats were orally treated with 0.3, 0.6, or 1.2 g·kg$^{-1}$ CMS-Na once a day for 7 d. Then, approximately 200 mg feces was collected and the pH was measured. Mean±SD, $n$ = 10, $^*P<0.05$,$^{**}P<0.01$, vs. normal.

the expression of Tph1 in the CMS-Na group was significantly higher (2.62±0.05 vs. 2.52±0.11 log (pg·mL$^{-1}$), $P<0.05$). Likewise, the expression of Tph1 in the CMS-Na + metronidazole group (2.33±0.22 log (pg·mL$^{-1}$)) and the CMS-Na + 4-chloro-dl-phenylalanine group (2.36 ±0.16 log (pg·mL$^{-1}$)) was significantly lower than that in the CMS-Na group ($P<0.05$ or $P<0.01$), suggesting that CMS-Na may be associated with the increase of SCFAs. The results of 5-HT level were similar to those for Tph1. Compared to the normal group (21.08±2.32μg/mL), the 5-HT content of CMS-Na group (23.30±2.27 μg·mL$^{-1}$) was significantly increased ($P<0.05$). The 5-HT content in the CMS-Na + metronidazole group (20.15±2.86 μg·mL$^{-1}$) and the CMS-Na+4-chloro-dl-phenylalanine group (18.89±2.20 μg·mL$^{-1}$) significantly decreased ($P<0.05$ or $P<0.01$) compared to the CMS-Na group. These results suggest that metronidazole and 4-chloro-dl- phenylalanine can inhibit the production of SCFAs.

## Discussion

Constipation is a common syndrome defined as a reduced bowel movements, a feeling of incomplete emptying and hard stools [15, 16]. In recent years, an increasing number of animal and human studies have shown that intestinal microbes can alleviate constipation symptoms by improving intestinal motility, which have shown promising results [17, 18]. Thus, alterations in the composition of the gut microbiota may lead to dysfunction of gut motility, which

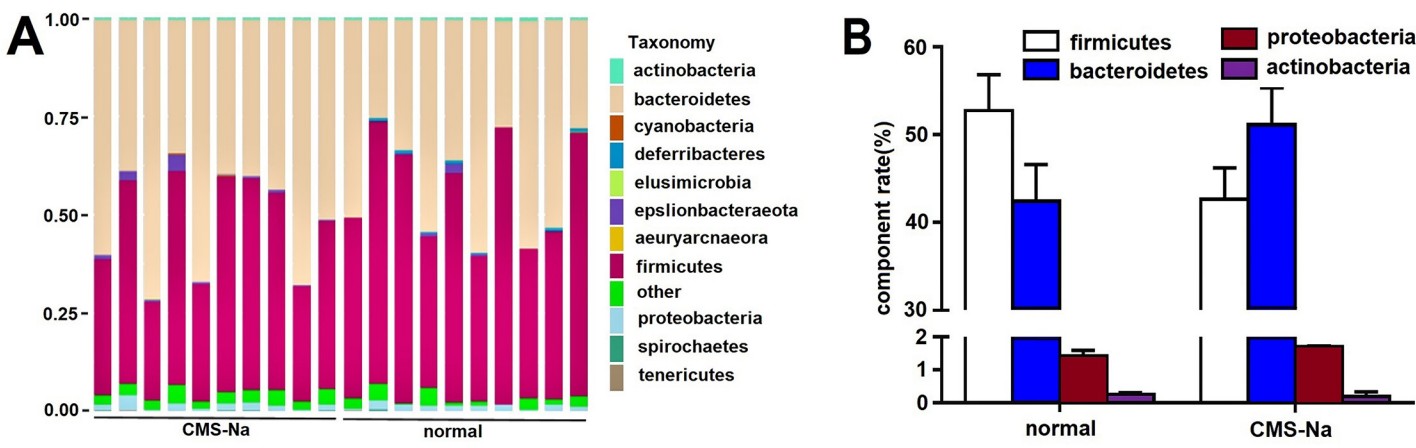

**Fig 5. Effect of CMS-Na on the content of Short-Chain Fatty Acids (SCFAs) in rat feces.** Rats were orally treated with 1.2 g·kg[-1] CMS-Na once a day for 5 d. Then, approximately 500 mg feces were collected to detect the contents of SCFAs by gas chromatography. Mean±SD, $n$ = 10, $^{*}P<0.05$, $^{**}P<0.01$, vs. normal.

**Fig 6. Effects of CMS-Na on different intestinal flora in rats.** (A) CMS-Na modulated the composition of the gut microbiota at different *phylum* levels. (B) Effect of CMS-Na on four different *phylum* levels. Mean ± SD, $n$ = 10.

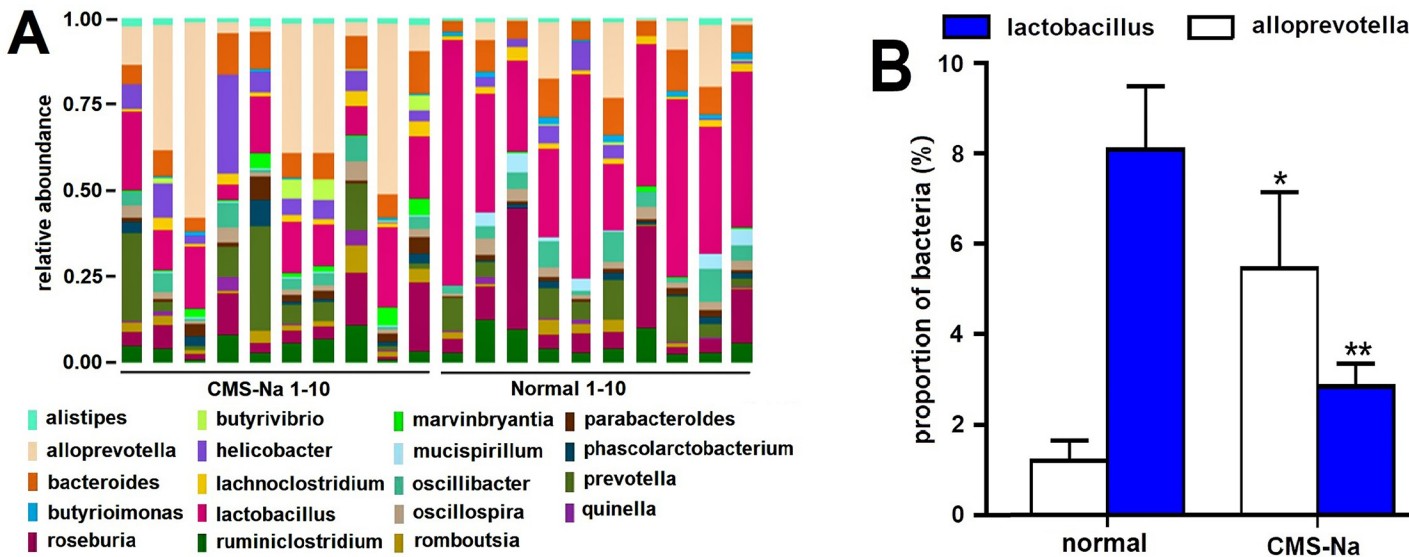

**Fig 7. Effect of CMS-Na on anaerobic bacteria.** (A) CMS-Na modulated the composition of the gut microbiota at different *genus* levels. (B) Effect of CMS-Na on two different anaerobic bacteria in the intestinal tracts of rats. Mean±SD, $n = 10$, $^*P < 0.05$, $^{**}P < 0.01$, vs. normal.

might be one of the pathogenic factors of constipation [19, 20]. Gut transit is the functional consequence of tonic and phasic gut contractions and refers to the time taken for the intraluminal contents to traverse the gastrointestinal tract [21]. Regulators of intestinal motility mainly include the central nervous system, enteric nervous system, immune system and gut luminal environment [7, 21]. The gut luminal environment involves the gastrointestinal microbiota and microbial fermentation products. The products of anaerobic bacteria fermentation, SCFAs, mainly comprise acetate, propionate, and butyrate [22].

SCFAs are end products of colonic anaerobe degradation of indigestible foods and complex carbohydrates, such as dietary fiber, prebiotics, or resistant starch [23–25]. SCFAs can change the pH of the colon, provide an energy source for colonic cells, regulate inflammatory responses [24, 26, 27], and affect metabolism, body weight, insulin sensitivity and cardiac metabolism [23, 26, 28]. Furthermore, it has been shown that SCFAs stimulate colonic transit via intraluminal 5-HT release in rats [29]. However, how SCFAs impact colonic motility remains unclear. Emerging studies have indicated that SCFAs are of major importance in improving gut motility through increasing serotonin synthesis in enterochromaffin cells (EC) [30]. This study showed that human- and mouse-derived gut microorganisms promote colonic Tph1 expression and 5-HT production through stimulatory activities of SCFAs on EC cells.

It is acknowledged that 5-HT is a neurotransmitter and autacoid with a variety of biological functions, including modulation of intestinal tract secretion and motility [31]. In the human body, 5-HT is synthesized from L-tryptophan by the action of the rate-limiting enzyme tryptophan hydroxylase and stored in secretory granules prior to release [32]. Tryptophan hydroxylase has two isomers, Tph1 and Tph2 [33]. Tph1 is mainly expressed in the skin, intestine, pineal gland and central nervous system, while Tph2 is only expressed in nerve cells [34, 35]. Latest data have shown that the vast majority of 5-HT (90%~95%) is produced by EC cells of the gut [36–38], where it is synthesized by Tph1. 5-HT secretion from EC cells can activate intrinsic primary afferent neurons, which initiates secretion and propulsive motility [38, 39].

Based on the above research background, we establish constipation model in mice by loperamide [40], which inhibits intestinal water secretion and delays colonic peristalsis [41, 42]. The

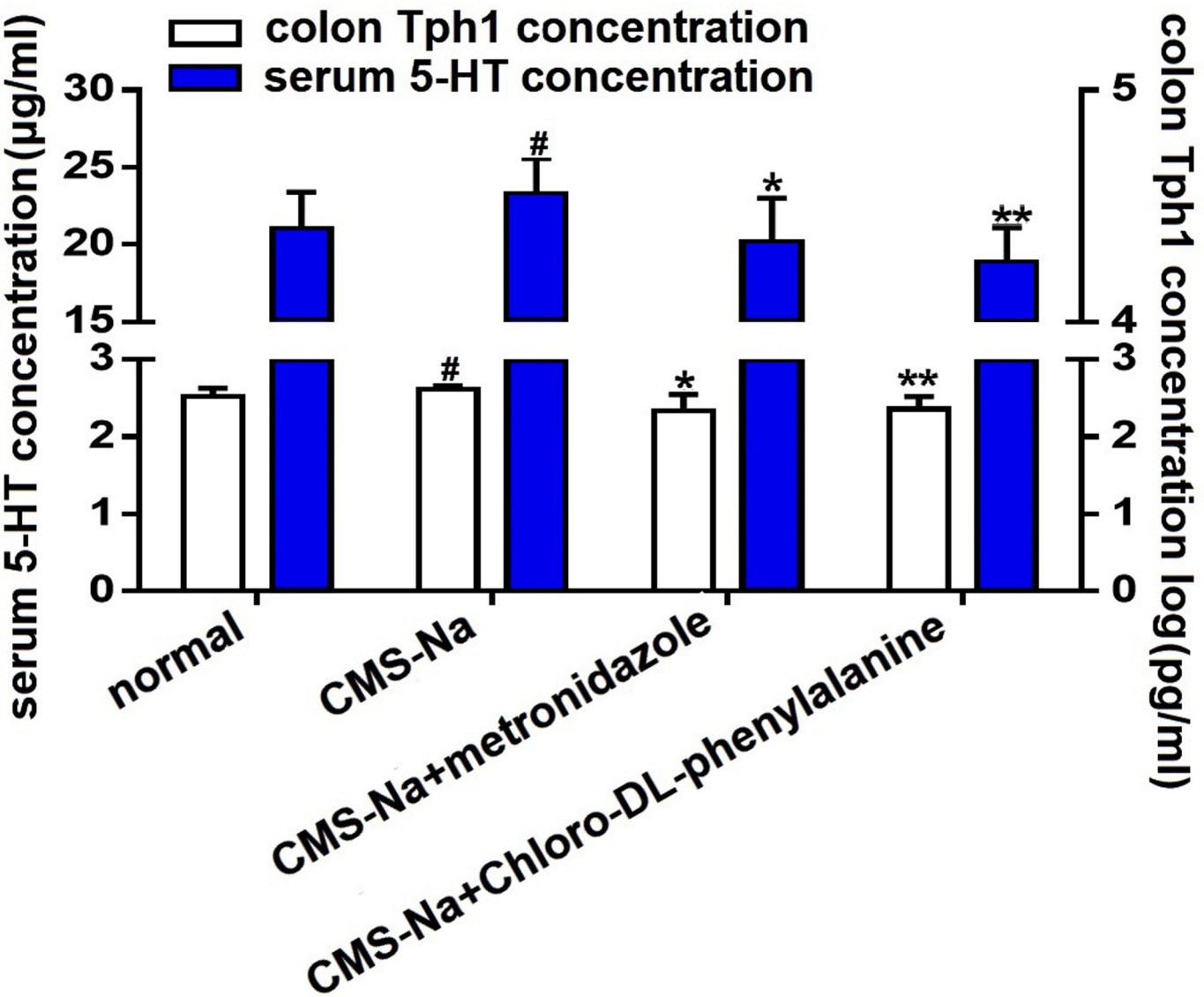

**Fig 8. Effect of CMS-Na on tryptophan hydroxylase-1(Thp1) in the colon and 5-hydroxytryptamine(5-HT) in the serum of rats.** Rats were orally treated with CMS-Na at a dose of 1.2 g·kg$^{-1}$ or vehicle once a day for 5 d. Rats blood were collected and the proximal colon (approximately 400 mg) was removed. Determination of Thp1 and 5-HT by ELISA. Mean±SD, $n = 10$, $^{*}P<0.05$, $^{**}P<0.01$, vs. CMS-Na group. $^{#}P<0.05$, vs. normal.

results showed that CMS-Na significantly increased the number of fecal pellets and the intestinal propulsive rate, suggesting that CMS-Na has a superior effect on relieving loperamide-induced constipation in mice. CMS-Na also improved the fecal water content and intestinal peristalsis, contributing to shortening the colonic transit time in normal rats.

It is reported that resistant starch influences production of SCFAs by fermentation, which results in a low pH environment in the intestinal tract and alters beneficial bacteria within the colon [43–46]. Similarly, our results showed that CMS-Na significantly decreased fecal pH and increased the contents of acetic acid, propionic acid and butyric acid. Furthermore, CMS-Na did not change the diversity of intestinal microorganisms in normal rats, but significantly increased the proportion of anaerobic *Alloprevotella* and decreased the proportion of *Lactobacillus* at the microbial *genus* level. The two kinds of bacteria both produced short-chain fatty

acids. Recent research showed that SCFAs increased the expression of Tph1, which is an important enzyme for synthesizing 5-HT in gut mucosal EC cells [30]. Our results showed that the levels of Tph1 in colon and 5-HT in serum were significantly lower when rats were treated with metronidazole and 4-chloro-dl-phenylalanine. Because metronidazole is an anaerobic inhibitor and 4-chloro-dl-phenylalanine is a tryptophan hydroxylase inhibitor, the results suggest that anaerobic and tryptophan hydroxylase are related to the production of SCFAs. Moreover, compared with the normal group, CMS-Na increased the levels of Tph1 and 5-HT, suggesting that the increase occurs through SCFAs.

## Conclusion

CMS-Na alleviates constipation. Meanwhile, CMS-Na regulates the proportion of the anaerobic bacteria *Alloprevotella* and *Lactobacillus* in the colon, increases short-chain fatty acids, and promotes the synthesis of Tph1 and 5-HT.

## Supporting information

**S1 Graphical abstract.**
(DOC)

## Author Contributions

**Conceptualization:** Wu-dang Lu, Yong-xiao Cao.

**Data curation:** Man-li Wu, Jun-xia Zhang.

**Formal analysis:** Man-li Wu, Jun-xia Zhang.

**Investigation:** Wu-dang Lu, Jun-xia Zhang, Ting-ting Huang, Shuai-shuai Du, Yong-xiao Cao.

**Methodology:** Wu-dang Lu, Man-li Wu.

**Project administration:** Man-li Wu, Yong-xiao Cao.

**Resources:** Man-li Wu, Yong-xiao Cao.

**Validation:** Wu-dang Lu, Man-li Wu, Yong-xiao Cao.

**Writing – original draft:** Wu-dang Lu, Man-li Wu.

**Writing – review & editing:** Wu-dang Lu, Yong-xiao Cao.

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
