## [Decision Letter · Decision Letter 0]

18 May 2021

PONE-D-21-07877

A study of sodium carboxymethyl starch with high substitution degree in defecation

PLOS ONE

Dear Dr. Cao,

Thank you for submitting your manuscript to PLOS ONE. After careful consideration, we feel that it has merit but does not fully meet PLOS ONE’s publication criteria as it currently stands. Therefore, we invite you to submit a revised version of the manuscript that addresses the points raised during the review process.

We look forward to receiving your revised manuscript.

Kind regards,

Zongxin Ling

Academic Editor

PLOS ONE

Journal Requirements:

2.Thank you for stating the following in the Financial Disclosure * (delete as necessary) section:

"This research was financially supported by Lipont Pharmaceutical Co. (China). The funders had no role in study design, data collection and analysis, decision to publish, or preparation of the manuscript."

We note that you received funding from a commercial source: Lipont Pharmaceutical Co.

b)Please state what role the funders took in the study.  If the funders had no role, please state: "The funders had no role in study design, data collection and analysis, decision to publish, or preparation of the manuscript."

Reviewers' comments:

Reviewer's Responses to Questions

**Comments to the Author**

1. Is the manuscript technically sound, and do the data support the conclusions?

Reviewer #1: Yes

Reviewer #2: Yes

2. Has the statistical analysis been performed appropriately and rigorously? 

Reviewer #1: I Don't Know

Reviewer #2: Yes

3. Have the authors made all data underlying the findings in their manuscript fully available?

Reviewer #1: Yes

Reviewer #2: Yes

4. Is the manuscript presented in an intelligible fashion and written in standard English?

Reviewer #1: Yes

Reviewer #2: No

5. Review Comments to the Author

Reviewer #1: I have not seen before such application of starch. seems interesting. Revision is suggested. Please see my important comments below

1-ToC graphical abstract could be added to improve the manuscript.

2-Figures (e.g., Fig. 3, 4, 5, 6, 8B): Please do not use Time new Roman font. Please use more readable fonts such as Helvetica and Arial. Please modify the figures throughout the manuscript. Also, some fonts may need to be enlarged to be more readable.

3-The color of fig 7 and 8 A are not visible well. The graph would be better to be colorful + having pattern. Therefore, it will be suitable for both B/W and colorful printing.

4- Fig 1 should be deleted. instead, A schematic illustration is needed on the reaction/preparation of the system to help the readers to understand the process.

5-References: There are some old references. Please remove/replace the outdated references. Some of them are listed below but please replace the other references.

Meijer-Severs GJ, Cats A, Verschueren RCJ, Santen EV, Kleibeuker JH. Anaerobes and their fermentation products in faeces of patients with familial adenomatous -

polyposis before and after subtotal colectomy and ileorectal anastomosis. European journal of clinical investigation. 1993 Jun; 23(6):356-60. https://doi.org/10.1111/j.1365-2362.1993.tb02036.x. PMID: 8393794. eng.

Richard K LL, Brown IL, Hu Y, Morita T, Esterman A, Young GP. Effect of dietary resistant starch and protein on colonic fermentation and intestinal tumourigenesis

in rats. Carcinogenesis. 2007 Feb; 28(2):240-245. https://doi.org/ 10.1093/carcin/bgl245 PMID: 17166881. Epub 2006/11/13.eng.

Englyst HN, Cummings JH. Digestion of the polysaccharides of some cereal foods in the human small intestine. Am J Clinic Nutr. 1985 Nov; 42(5):778-787.

https://doi.org/10.1093/ajcn/42.5.778 PMID: 2998174. Epub 1985/11/01.eng.

Meijer-Severs GJ, Cats A, Verschueren RC, Santen EV, Kleibeuker JH. Anaerobes and their fermentation products in faeces of patients with familial adenomatous polyposis before and after subtotal colectomy and ileorectal anastomosis. Europe J Clinic Invest. 1993 Jun; 23(6):356-60. https://doi.org /10.1111/j.1365-

2362.1993.tb02036.x PMID: 8393794.eng

6-There are many errors in the references.

7-Introductio needs to be revised carfully. It was not written well, the aim of it is not apparent. What is the problem and potential disease of defecation? It should be added. Also no need of use abbreviation of Resistant starch (RS). updated ref should be added into introduction, e.g., Maspes et al, Gels 7 (1), 6, 2021;; Makvandi et al, Environmental Chemistry Letters, 1-29, 2021;;

8-Conc,usion should be elaborated.

Reviewer #2: This paper describes the effect and potential mechanism of CMS Na in prevention and treatment. The main problem is to clarify the relationship between the causes and results of different animal models. At the same time, it needs further improvement in writing and grammar.

1. Page 1, the title of this manuscript should more accurately reflect the core content of the research.

2. Page 2, “Food additive, Sodium carboxymethyl starch” should be “Food additive, sodium carboxymethyl starch”. “gas chromatogra- phy” should be “gas chromatogra- phy”. “the proportion of Alloprevotella” should be “the relative abundance of Alloprevotella”, so were those in the other parts of this manuscript. Which kind of the biodiversity should be pointed out.

3. Page 2, from the current data, the author cannot get the conclusion “by regulating Alloprevotella and Lactobacillus in the colon, increasing short-chain fatty acids and promoting the synthesis of Tph1 and 5-HT”. The mechanism may be only related to the above observations.

4. Page 4, Figure 1 should be integrated into other figures. Figure 2 is unscientific and should be deleted.

5. In the method and results section, why did the author build the same model with rats and mice? It is more important to clarify why different experiments of the same study, such as water content, fecal pH, short chain fast acid, 16S rDNA sequence analysis TPH1 and 5-HT in serum, were performed in rats or mice, rather than on the same animal. This makes the whole article more confusing, and it is difficult to analyze the correlation between the results.

6. In the whole result section, the author should make the result title more accurately reflect the important content of each section, rather than a general description.

7. Page 10, The following description is not clear. “The first 2,000,000 reads of the document were counted. Reads with a length > 75 BPwere truncated to 50 BP for statistical duplication.” It is suggested to modify the description and put it into the method. In addition, it is better to describe how many samples have been tested and how many clean reads each sample has.

8. Page 11, if there is no significant change at the phylum level, it is unnecessary to describe. It is suggested to add the description of the changes of the Family level. The increase or decrease should be described as those of the relative abundance of a particular bacterial taxonomy. Family, genes should be italicized.

9. Page 15, this conclusion is not accurate. The effect of CMS Na alleviates con stitution is only related to that described later such as regulating the proportion of the anaerobic bacteria Alloprevotella and Lactobacillus in the colon, increasing short-chain fatty acids, and promoting the synthesis of Tph1 and 5-HT., and its causality has not been verified.

10. In the figures，such figure 5 and 10, don't mark A when there is only one panel.

6. PLOS authors have the option to publish the peer review history of their article (what does this mean?). If published, this will include your full peer review and any attached files.

Reviewer #1: No

Reviewer #2: No

---

## [Author Response · Author response to Decision Letter 0]

25 Jun 2021

Dear Zongxin editor, 

Thank you very much for your kind consideration for evaluation of our research article PONE-D-21-07877: The effect of carboxymethyl starch sodium with high substitution degree on defecation. Your comments are constructive and benefit for our article. Based on your suggestions, we have modified the manuscript as following table. And a revised manuscript is attached. We want it can meet your requirements.

In addition, the research was financially supported by Lipont Pharmaceutical Co. a) we stated that the study is not related to employment, consultancy, patents, products in development, marketed products of supporter. b) The funders had no role in study design, data collection and analysis, decision to publish, or preparation of the manuscript. we declare that there is no competing conflict of interest. This does not alter our adherence to PLOS ONE policies on sharing data and materials.

We look forward to receiving your reply.

Best Regards,

Yong-xiao Cao

Department of Pharmacology

Xi`an Jiaotong University Health Science Center

yxy@xjtu.edu.cn

Reviewer’s Comments Authors’ Response

Reviewer #1 /

Comment 1: ToC graphical abstract could be added to improve the manuscript.

Thank you for the suggestion. Graphical abstract had been added as separate document.

Comment 2: 

Figures (e.g., Fig. 3, 4, 5, 6, 8B): Please do not use Time new Roman font. Please use more readable fonts such as Helvetica and Arial. Please modify the figures throughout the manuscript. Also, some fonts may need to be enlarged to be more readable. 

Thank your suggestion. Time new Roman font has been replaced by Arial in Fig. 3, 4, 5, 6, 8B, some fonts has be enlarged in Fig. 8, 9, Fig. 7 has be deleted.

Comment 3: 

The color of fig 7 and 8 A are not visible well. The graph would be better to be colorful + having pattern. Therefore, it will be suitable for both B/W and colorful printing. 

Thanks you for the comment. Fig 7 has been deleted and the color of Fig 8 has been changed.

Comment 4: 

Fig 1 should be deleted. instead, A schematic illustration is needed on the reaction/preparation of the system to help the readers to understand the process. 

Thanks you. Fig 1 has be deleted. A schematic illustration of reaction process for the preparation of sodium carboxymethyl starch has been expressed to help the readers to understand the preparation process of sodium carboxymethyl starch. 

Comment 5: 

References: There are some old references. Please remove/replace the outdated references. Some of them are listed below but please replace the other references.

Meijer-Severs GJ, Cats A, Verschueren RCJ, Santen EV, Kleibeuker JH. Anaerobes and their fermentation products in faeces of patients with familial adenomatous polyposis before and after subtotal colectomy and ileorectal anastomosis. European journal of clinical investigation. 1993 Jun; 23(6):356-60. https://doi.org /10.1111/j.1365-2362.1993.tb02036.x. PMID: 8393794. eng.

Richard K LL, Brown IL, Hu Y, Morita T, Esterman A, Young GP. Effect of dietary resistant starch and protein on colonic fermentation and intestinal tumourigenesis in rats. Carcinogenesis. 2007 Feb; 28(2):240-245. https:// doi. org/10.1093/carcin/bgl245 PMID: 17166881. Epub 2006/11/13.eng.

Englyst HN, Cummings JH. Digestion of the polysaccharides of some cereal foods in the human small intestine. Am J Clinic Nutr. 1985 Nov; 42(5): 778- 787. https://doi.org/10.1093/ ajcn/ 42.5.778 PMID: 2998174. Epub 1985/ 11/01.eng. 

Thank you for the suggestion. The references have been updated.

Comment 6: 

There are many errors in the references. 

Thanks you. All of references have been revised in accordance with PLOS ONE guidelines.

Comment 7: 

Introduction needs to be revised carfully. It was not written well, the aim of it is not apparent. What is the problem and potential disease of defecation? It should be added. Also no need of use abbreviation of Resistant starch (RS). updated ref should be added into introduction, e.g., Maspes et al, Gels 7 (1), 6, 2021;; Makvandi et al, Environmental Chemistry Letters, 1-29, 2021; 

Thanks you for the comment. The introduction has been rewritten.The aim, problem and potential disease of defecation have been descripted. Abbreviation, RS has been replacrd by resistant starch. The corresponding references were updated.

Comment 8: 

Conclusion should be elaborated. 

Thanks you for the comment. The conclusion section has been modifyied as CMS-Na alleviates constipation, and the mechanism may be related to regulating the proportion of the anaerobic bacteria Alloprevotella and Lactobacillus in the colon, increasing short-chain fatty acids, and promoting the synthesis of Tph1 and 5-HT.

Reviewer #2: 

Comment 1: 

Page 1, the title of this manuscript should more accurately reflect the core content of the research. 

Thank your suggestion. The title has been modified as “the effect of carboxymethyl starch sodium with high substitution degree on defecation”.

Comment 2: 

Page 2,“Food additive, Sodium carboxymethyl starch” should be “Food additive, sodium carboxymethyl starch”. “gas chromatogra- phy” should be “gas chromatogra-phy”. “the proportion of Alloprevotella” should be “the relative abundance of Alloprevotella”, so were those in the other parts of this manuscript. Which kind of the biodiversity should be pointed out. 

Thanks you for the comment.

1.“Food additive, Sodium carboxymethyl starch” has been modified “Food additive, sodium carboxymethyl starch”.

2.“gas chromatogra- phy” has been modified“gas chromatography”. 

3.“the proportion of Alloprevotella” has been modified“the relative abundance of Alloprevotella”.

Other parts of the manuscript have been revised in detail.

Comment 3: 

Page 2, from the current data, the author cannot get the conclusion “by regulating Alloprevotella and Lactobacillus in the colon, increasing short-chain fatty acids and promoting the synthesis of Tph1 and 5-HT”. The mechanism may be only related to the above observations. 

We agree with your comments. The conclusion has been modified as: CMS-Na promotes defecation of constipated mice. The underlying mechanism may be related to regulating Alloprevotella and Lactobacillus in the colon, increasing short-chain fatty acids and promoting the synthesis of Tph1 and 5-HT.

Comment 4: 

Page 4, Figure 1 should be integrated into other figures. Figure 2 is unscientific and should be deleted. 

Thank you for the comments. 

Fig 1 has been modified. Fig 2 has been deleted.

Comment 5: 

In the method and results section, why did the author build the same model with rats and mice? It is more important to clarify why different experiments of the same study, such as water content, fecal pH, short chain fast acid, 16S rDNA sequence analysis TPH1 and 5-HT in serum, were performed in rats or mice, rather than on the same animal. This makes the whole article more confusing, and it is difficult to analyze the correlation between the results. 

Thanks you for the comment. The models with rats and mice are different.

1.Mice were orally administered loperamide to prepare a constipation model. Defecation granules and intestinal propulsion rate were measured. The results suggest that CMS-Na has the effect of relieving constipation.

2. Rats were directly gavaged with different doses of CMS-Na. and rat constipation model was not established. The indicators included: fecal water content, fecal pH, short chain fast acid, 16S rDNA sequence analysis TPH1 and 5-HT in serum. 

Comment 6: 

In the whole result section, the author should make the result title more accurately reflect the important content of each section, rather than a general description. 

Thanks you for the comment. 

In the whole result section, result titles has been added to more accurately reflect the important content of each section.

Comment 7: 

Page 10, The following description is not clear. “The first 2,000,000 reads of the document were counted. Reads with a length >75 BP were truncated to 50 BP for statistical duplication.” It is suggested to modify the description and put it into the method. In addition, it is better to describe how many samples have been tested and how many clean reads each sample has. 

Thank you for the comment.

The sentences, “The first 2,000, 000 reads of the document were counted. Reads with a length > 75 BP were truncated to 50 BP for statistical duplication.”has been deleted.

In addition, the numbers of sample has been documented in the method section.

Comment 8: 

Page 11, if there is no significant change at the phylum level, it is unnecessary to describe. It is suggested to add the description of the changes of the Family level. The increase or decrease should be described as those of the relative abundance of a particular bacterial taxonomy. Family, genes should be italicized. 

I agree with your suggestion that the description of the results has been simplified, in addition, Family, genes had been italicized.

Comment 9: 

Page 15, this conclusion is not accurate. The effect of CMS-Na alleviates con stitution is only related to that described later such as regulating the proportion of the anaerobic bacteria Alloprevotella and Lactobacillus in the colon, increasing short-chain fatty acids, and promoting the synthesis of Tph1 and 5-HT, and its causality has not been verified. 

I agree to your suggestion. The conclusion has been mofified as CMS-Na alleviates constipation, and the mechanisms may be related to regulating the proportion of the anaerobic bacteria Alloprevotella and Lactobacillus in the colon, increasing short-chain fatty acids, and promoting the synthesis of Tph1 and 5-HT.

Comment 10: 

In the figures，such figure 5 and 10, don't mark A when there is only one panel. 

Thank you for the comment. Mark A/B in figure 5 and 10 have been deleted.

Competing Interests Statement

The research was financially supported by Lipont Pharmaceutical Co. 

a)we stated that the study is not related to employment, consultancy, patents, products in development, marketed products of supporter.

b)The funders had no role in study design, data collection and analysis, decision to publish, or preparation of the manuscript.

In addition, we declare that there is no competing conflict of interest. This does not alter our adherence to PLOS ONE policies on sharing data and materials.

---

## [Decision Letter · Decision Letter 1]

30 Jun 2021

PONE-D-21-07877R1

The effect of carboxymethyl starch sodium with high substitution degree on defecation

PLOS ONE

Dear Dr. Cao,

Thank you for submitting your manuscript to PLOS ONE. After careful consideration, we feel that it has merit but does not fully meet PLOS ONE’s publication criteria as it currently stands. Therefore, we invite you to submit a revised version of the manuscript that addresses the points raised during the review process.

We look forward to receiving your revised manuscript.

Kind regards,

Zongxin Ling

Academic Editor

PLOS ONE

Journal Requirements:

Reviewers' comments:

Reviewer's Responses to Questions

**Comments to the Author**

1. If the authors have adequately addressed your comments raised in a previous round of review and you feel that this manuscript is now acceptable for publication, you may indicate that here to bypass the “Comments to the Author” section, enter your conflict of interest statement in the “Confidential to Editor” section, and submit your "Accept" recommendation.

Reviewer #1: (No Response)

Reviewer #2: All comments have been addressed

2. Is the manuscript technically sound, and do the data support the conclusions?

Reviewer #1: (No Response)

Reviewer #2: Yes

3. Has the statistical analysis been performed appropriately and rigorously? 

Reviewer #1: (No Response)

Reviewer #2: Yes

4. Have the authors made all data underlying the findings in their manuscript fully available?

Reviewer #1: (No Response)

Reviewer #2: Yes

5. Is the manuscript presented in an intelligible fashion and written in standard English?

Reviewer #1: (No Response)

Reviewer #2: No

6. Review Comments to the Author

Reviewer #1: The quality of the figures seems low. Some fonts are too big or in some case small.

The authors claimed to address the references, but it seems that the references were not implemented.

Reviewer #2: The authors have adequately addressed our comments raised in a previous round of review. However, before be published, this manuscript should be carefully checked for grammatical errors. Furthermore, the clarity of all figures needs to be improved. Usually, authors should provide pictures (300DPI) in tiff format.

7. PLOS authors have the option to publish the peer review history of their article (what does this mean?). If published, this will include your full peer review and any attached files.

Reviewer #1: No

Reviewer #2: No

---

## [Author Response · Author response to Decision Letter 1]

1 Aug 2021

Dear Zongxin, 

Thank you for your kind evaluation of our research article PONE-D-21-07877: The effect of sodium carboxymethyl starch with high degree of substitution on defecation. Based on your constructive comments, we have modified our manuscript as shown in the following table. Meanwhile, highpoint and clean manuscripts are also attached. We hope the revised version now will meet your requirements. 

In addition, the research was financially supported by Lipont Pharmaceutical Co. a) we stated that the study is not related to employment, consultancy, patents, products in development, marketed products of supporter. b) The funders had no role in study design, data collection and analysis, decision to publish, or preparation of the manuscript. we declare that there is no competing conflict of interest. This does not alter our adherence to PLOS ONE policies on sharing data and materials.

We look forward to receiving your reply.

Best Regards,

Yong-xiao Cao

Department of Pharmacology

Xi`an Jiaotong University Health Science Center

yxy@xjtu.edu.cn

---

## [Decision Letter · Decision Letter 2]

23 Aug 2021

The effect of sodium carboxymethyl starch with high degree of substitution on defecation

PONE-D-21-07877R2

Dear Dr. Cao,

We’re pleased to inform you that your manuscript has been judged scientifically suitable for publication and will be formally accepted for publication once it meets all outstanding technical requirements.

Kind regards,

Zongxin Ling

Academic Editor

PLOS ONE

Additional Editor Comments (optional):

Reviewers' comments:

Reviewer's Responses to Questions

**Comments to the Author**

1. If the authors have adequately addressed your comments raised in a previous round of review and you feel that this manuscript is now acceptable for publication, you may indicate that here to bypass the “Comments to the Author” section, enter your conflict of interest statement in the “Confidential to Editor” section, and submit your "Accept" recommendation.

Reviewer #1: (No Response)

Reviewer #2: All comments have been addressed

2. Is the manuscript technically sound, and do the data support the conclusions?

Reviewer #1: (No Response)

Reviewer #2: Yes

3. Has the statistical analysis been performed appropriately and rigorously? 

Reviewer #1: (No Response)

Reviewer #2: Yes

4. Have the authors made all data underlying the findings in their manuscript fully available?

Reviewer #1: (No Response)

Reviewer #2: Yes

5. Is the manuscript presented in an intelligible fashion and written in standard English?

Reviewer #1: (No Response)

Reviewer #2: Yes

6. Review Comments to the Author

Reviewer #1: (No Response)

Reviewer #2: The author has made detailed modifications to the paper. Before publishing, you need to carefully calibrate full text. For example, the “aim” inside the summary is written as “arm”.

7. PLOS authors have the option to publish the peer review history of their article (what does this mean?). If published, this will include your full peer review and any attached files.

Reviewer #1: **Yes: **Pooyan Makvandi

Reviewer #2: No

---

## [Editor Report · Acceptance letter]

26 Aug 2021

PONE-D-21-07877R2 

The effect of sodium carboxymethyl starch with high degree of substitution on defecation 

Dear Dr. Cao:

I'm pleased to inform you that your manuscript has been deemed suitable for publication in PLOS ONE. Congratulations! Your manuscript is now with our production department. 

Kind regards, 

on behalf of

Dr. Zongxin Ling 

Academic Editor

PLOS ONE